Integrated lipidomics and RNA sequencing analysis reveal novel changes during 3T3-L1 cell adipogenesis

Pei Yangli 1
Song Yuxin 1
Wang Bingyuan 2
Lin Chenghong 1
Yang Ying 1
Li Hua 1
Feng Zheng greatfz@126.com 1
1 Guangdong Provincial Key Laboratory of Animal Molecular Design and Precise Breeding, School of Life Science and Engineering, Foshan University , Foshan , China
2 Institute of Animal Sciences, Chinese Academy of Agricultural Sciences , Beijing , China
Haraguchi Tokuko
Electronic publication date: 2022 May 3
Publication date: 2022
Volume: 10
Electronic Location ID: e13417
Received 2022 Jan 21; Accepted 2022 Apr 19
Copyright: ©2022 Pei et al.
Copyright year: 2022
Copyright holder: Pei et al.
License: This is an open access article distributed under the terms of the Creative Commons Attribution License, which permits unrestricted use, distribution, reproduction and adaptation in any medium and for any purpose provided that it is properly attributed. For attribution, the original author(s), title, publication source (PeerJ) and either DOI or URL of the article must be cited.
License URL: https://creativecommons.org/licenses/by/4.0/

Keywords: Adipogenesis, Lipidomics, 3T3-L1, Triacylglycerol, Diacylglycerol, Sphingomyelin, Ceramide, Phospholipid

Funding: The National Science Foundation for Young Scientists of China 31702088 The Guangdong Provincial Key Laboratory of Animal Molecular Design and Precise Breeding 2019B030301010 The Key Laboratory of Animal Molecular Design and Precise Breeding of Guangdong Higher Education Institutes 2019KSYS011 The Foshan University Initiative Scientific Research Program This work was supported by the National Science Foundation for Young Scientists of China (31702088), the Guangdong Provincial Key Laboratory of Animal Molecular Design and Precise Breeding (2019B030301010), the Key Laboratory of Animal Molecular Design and Precise Breeding of Guangdong Higher Education Institutes (2019KSYS011), and the Foshan University Initiative Scientific Research Program. The funders had no role in study design, data collection and analysis, decision to publish, or preparation of the manuscript.

==============================
After adipogenic differentiation, key regulators of adipogenesis are stimulated and cells begin to accumulate lipids. To identify specific changes in lipid composition and gene expression patterns during 3T3-L1 cell adipogenesis, we carried out lipidomics and RNA sequencing analysis of undifferentiated and differentiated 3T3-L1 cells. The analysis revealed significant changes in lipid content and gene expression patterns during adipogenesis. Slc2a4 was up-regulated, which may enhance glucose transport; Gpat3, Agpat2, Lipin1 and Dgat were also up-regulated, potentially to enrich intracellular triacylglycerol (TG). Increased expression levels of Pnpla2, Lipe, Acsl1 and Lpl likely increase intracellular free fatty acids, which can then be used for subsequent synthesis of other lipids, such as sphingomyelin (SM) and ceramide (Cer). Enriched intracellular diacylglycerol (DG) can also provide more raw materials for the synthesis of phosphatidylinositol (PI), phosphatidylcholine (PC), phosphatidylethanolamine (PE), ether-PE, and ether-PC, whereas high expression of Pla3 may enhance the formation of lysophophatidylcholine (LPC) and lysophosphatidylethanolamine (LPE). Therefore, in the process of adipogenesis of 3T3-L1 cells, a series of genes are activated, resulting in large changes in the contents of various lipid metabolites in the cells, especially TG, DG, SM, Cer, PI, PC, PE, etherPE, etherPC, LPC and LPE. These findings provide a theoretical basis for our understanding the pathophysiology of obesity.

Introduction

Adipogenesis is a multi-step process regulated by a complex signaling network, resulting in dramatic changes in cell morphology (Jakab et al., 2021). After adipogenesis, mature adipocytes are occupied by large lipid droplets (Wang et al., 2013). Mouse 3T3-L1 preadipocytes can be used as a model for studying adipogenesis and a medium containing insulin, dexamethasone (Dex), and methylisobutylxanthine (IBMX) can induce the preadipocytes to differentiate. Insulin, a stimulator for insulin-like growth factor 1 (IGF-1), is critically important for adipogenic differentiation (Haeusler, McGraw & Accili, 2018). Dex, an anti-inflammatory 9-fluoro-glucocorticoid, can inhibit proliferation of adipocytes, and also enhance the activity of several transcription factors required for differentiation, including members of the C/EBPs family, promoting terminal differentiation of adipocytes (Shugart & Umek, 1997; Tomlinson et al., 2010). IBMX, a phosphodiesterase inhibitor, acts as a cAMP inducer and activates protein kinase A (PKA) to promote preadipocyte differentiation into adipocytes (Farmer, 2006).

Several days after induction in vitro, fibroblast-like cells develop into round adipocyte-like cells and begin to accumulate lipid droplets (LDs). Final differentiation occurs after seven to ten days of induction (Gabrielli et al., 2018). LDs consist of a lipid ester nucleus wrapped by a phospholipid monolayer and they are most conspicuous in white adipocytes, which have a large single-compartment lipid storage function (Fujimoto & Parton, 2011). Due to the prevalence of obesity, type 2 diabetes, and metabolic syndrome worldwide, regulation of lipid storage and utilization has become the focus of many studies. It is crucial to understand the changes and mechanisms of adipogenesis to develop therapeutic strategies for these diseases.

Large numbers of LDs accumulate in mature adipocytes. However, the species of enriched lipids, and how the lipid profile changes over the course of adipogenesis, are still unknown. Therefore, in the current study, we combined lipidomics and RNA sequencing to analyze differences in lipid content and lipid-associated genes between undifferentiated and differentiated 3T3-L1 cells.

Material and Methods

Cell culture and 3T3-L1 cell differentiation

3T3-L1 cells were obtained from Prof. Shulin Yang (Institute of Animal Sciences, Chinese Academy of Agricultural Sciences). Dulbecco’s Modified Eagle’s Medium (DMEM) (11965; Gibco) supplemented with 10% FBS (12483, Gibco) and 1% penicillin-streptomycin (15240; Gibco) (maintaining medium) was used to culture 3T3-L1 cells. Cultures were conducted with 1 ×106 cells/well in 6-well cell culture plates. These cells were grown until confluence (day 0), then differentiation was induced by adding 1 mM insulin (I5523; Sigma Aldrich), 0.25 mM dexamethasone (D175; Sigma Aldrich), and 0.5 mM 3-isobutyl-1- methylxanthine (15879, Sigma Aldrich) in the maintaining medium. After 4 days, cells were cultured in maintaining medium with 1 mM insulin. On day 10, the fully differentiated adipocytes were used for Oil Red O (O1391; Sigma Aldrich) staining. Cells at day 0 (D0) and day 10 (D10) were collected and stored at −80 °C for subsequent experiments.

Nile Red O Staining

Cells were treated with 4% PFA for 10 min. Afterwards, cells were washed with PBS and incubated with Nile Red O (O1391; Sigma Aldrich) for 30 min. After staining, images were acquired using a LEICA DMi8 microscope.

Lipid extraction and lipidomics study

Collected cells were thawed on ice, and lipids were extracted with isopropanol (IPA). A total of 120 µL precooled IPA was added, then samples were vortexed for 1 min, and incubated for 10 min at room temperature. The extraction mixture was then stored at −20 °C overnight. After centrifugation at 4,000 g for 20 min, the supernatants (one per specimen) were diluted 1:10 with IPA/acetonitrile (ACN)/H2O (2:1:1, v:v:v) and stored at −80 °C before analysis. A total of 10 µL of each extraction mixture was removed to prepare the pooled QC samples.

All lipid samples were analyzed by liquid chromatography–mass spectrometry (LC-MS) using an ultra-high performance liquid chromatography (UPLC) system (SCIEX, UK) and a high-resolution tandem mass spectrometer TripleTOF5600plus (SCIEX, UK). The acquired LC–MS raw data were analyzed by XCMS software (SCIEX, Warrington, UK), and the retention time (RT) and M/Z data were used to identify each ion. The online Kyoto Encyclopedia of Genes and Genomes (KEGG), Human Metabolome Database (HMDB), and in-house databases were used to perform level-one and level-two identification and annotation. MS2 indicates the metabolites that not only matched with level-one fragment ions, but also matched with level-two fragment ions in in-house database. Lipid metabolites with VIP (variable importance in projection) ≥1 and fold change > 2 or < 2 were considered statistically significantly different (Li et al., 2018).

mRNA library construction and sequencing

We used TRIzol reagent (Invitrogen, Waltham, MA) to exact total RNA from cells in accordance with the manufacturer’s procedure. Then we analyzed the RNA quantity and purity. Poly-T oligo-attached magnetic beads were used to purify Poly(A) RNA from total RNA with two rounds of purification. After purification, the mRNA was divided into small pieces at a high temperature using divalent cations. Then, according to the procedures of the mRNA Seq sample preparation kit, the cut RNA fragments were reverse transcribed to generate the final cDNA library (Illumina, San Diego, CA, USA). FastQC (version 0.11.2) was used to evaluate the quality of sequenced data. HISAT2 (version 2.0.4) was used to get clean data compared to the Mus musculus genome (Ensembl v101) (Kim, Langmead & Salzberg, 2015). StringTie (version 1.3.4) and Gffcompare (version 0.9.8) were used to assemble and quantify the transcripts, respectively (Pertea et al., 2015; Pertea et al., 2016). Differential gene expression between two groups (three biological replicates per condition) was performed using the DESeq2 R package available from Bioconductor (Love, Huber & Anders, 2014). Genes with an adjusted p-value < 0.05 and fold change > 2 or < −2 found by DESeq were assigned as differentially expressed. The statistical power, calculated using RNASeqPower (https://doi.org/doi:10.18129/B9.bioc.RNASeqPower) was 0.9829191. RNA sequencing data can be accessed on the SRA database, using accession number PRJNA795061.

Quantitative real time PCR (qRT-PCR)

qRT-qPCR was performed using TaKaRa SYBR Premix EX Taq (TaKaRa RR420A, JAPAN) on a QuantStudio5 Real-Time PCR System (Applied Biosystems, Waltham, MA, USA). All experiments contained three biological replicates, and each sample was quantified in duplicate. SPSS software and Excel were used to analyze the data. Relative gene expression was analyzed as 2−ΔΔCt, and students t -test was used to calculate statistical difference. Data was shown as mean ± standard error (SE). Primer sequences are shown in Table S1.

Correlation analysis

Three sets of transcriptome data and three sets of lipidomics data were used for correlation analysis. Correlation analysis was performed using the OmicStudio tools at https://www.omicstudio.cn/tool.

Results

3T3-L1 adipogenesis

Oil Red O staining showed that accumulation of lipid droplets within the cells increased during induction of differentiation (Figs. 1A–1F). In order to identify the changes between undifferentiated and differentiated 3T3-L1 cells, we collected samples at D0 and D10 for subsequent testing.

LC-MS-based untargeted lipidomics

We analyzed a total of 12 3T3-L1 preadipocyte and mature adipocyte samples (n = 6 per group) by LC-MS-based untargeted lipidomics to reveal differences in lipid composition. A total of 5,335 features were detected in positive ion mode, of which 3,187 could be annotated, with 850 matches to our in-house database. A total of 1,554 features were detected in negative anion mode, 873 of which could be annotated, with 378 matches to the in-house database. A total of 1,228 MS2 metabolites were identified from positive and negative ion modes (Table S2). Among all of the detected features, there were more up-regulated metabolites than down-regulated metabolites in mature fat cells (Fig. 2A).

Lipid profiles were compared using principal component analysis (PCA) (Fig. 2B). In unsupervised mode, the samples clustered together by cell type. Partial least squares discriminant analysis (PLS-DA) was used to identify the altered metabolites, and we found significant differences between the D0 and D10 groups (Figs. 2C, 2D). These results demonstrated significant lipid metabolite changes during differentiation of 3T3-L1 cells.

Figure 1 Oil Red O staining of 3T3-L1 cells during adipogenesis.

Oil Red O staining of 3T3-L1 cells at D0 (A), D2 (B), D4 (C), D6 (D), D8 (E), and D10 (F) after duction. The red dots inside the cells are lipid droplets.

Figure 2 Volcano plot, PCA and PLS-DA of the detected compounds in the two groups.

(A) Volcano plot of lipid metabolites. (B) PCA scatter plot of differentially expressed lipid metabolites. (C) PLS-DA score plots of D0 and D10 cells based on the extracted spectral data. (D) Permutation plot of PLS-DA based on the extracted spectral data.

The q-values obtained by Benjamini–Hochberg (BH) correction using univariate fold-change analysis, the t-test, and the VIP value obtained by PLS-DA were analyzed with multivariate statistics and used to screen for differentially expressed lipid metabolites. Based on VIP values and relative abundance, we identified 454 differentially expressed MS2 metabolites between D10 and D0, among which 301 were up-regulated and 153 were down-regulated (Fig. 3A). These differentially expressed metabolites included 214 glycerophospholipids (GPs), 152 glycerolipids (GLs), 73 sphingolipids (SPs), ten fatty acyls (FAs), three sterol lipids (ST), and two prenol lipids (PLs) (Fig. 3B, Table S3). Among the 214 differentially expressed GPs, there were 50 PCs (26 decreased, 24 increased), 36 LPCs (16 decreased, 20 increased), 29 EtherPEs (17 decreased, 12 increased), 23 PEs (4 decreased, 19 increased), 17 PIs (one decreased, 16 increased), 11 EtherPCs (eight decreased, three increased), 11 LNAPEs (one decreased, 10 increased) and nine LPEs (two decreased, seven increased). Among the 152 differentially abundant GLs, there were 66 TGs (two decreased, 64 increased), 27 EtherTGs (four decreased, 23 increased), 25 DGs (all increased), and 23 EtherMGDGs (18 decreased, five increased). The 73 differentially expressed SPs included 28 SMs (16 decreased, 12 increased), 12 Cer_NS (three decreased, nine increased), nine HexCer_NS (4 decreased, five increased) and eight SHexCer (5 decreased, three increased).

Figure 3 Differentially expressed lipid metabolites in two groups.

(A) Among 454 identified differentially expressed metabolites, 301 were upregulated and 153 were downregulated. (B) The differentially expressed lipid metabolites included 214 glycerophospholipids (GPs), 152 glycerolipids (GLs), 73 sphingolipids (SPs), 10 fatty acyls (FAs), three sterol lipids (STs) and two prenol lipids (PLs).

The top 10 down-regulated lipids were PS 18:0_22:6, LPC 30:0-SN1, CAR 18:1, CAR 16:0, PE O-18:2_22:5, LPC 36:2-SN1, PC O-18:1_18:1, LDGCC 40:6, PE 38:4, and SMGDG O-11:0_28:6. The top 10 up-regulated lipids were all TGs: TG 16:1_16:1_17:1, TG 15:0_16:0_16:1, TG 14:1_16:1_17:1, TG 10:0_16:0_16:1, TG 16:0_16:1_16:2, TG 14:1_16:1_16:1, TG 14:0_16:1_16:1, TG 14:0_15:0_16:1, TG 8:0_16:0_16:1, and TG 14:0_14:1_16:1 (Table 1 and Table S3).

Table 1 The top 10 down-regulated and up-regulated lipids in D10 versus D0 groups.

Metabolite	ratio	t.test_p.value	VIP	regulated	MS2class	MS2kegg	
TG 14:0_14:1_16:1	1099.99	1.03E−10	2.47326	up	TG	C00422	
TG 8:0_16:0_16:1	919.616	8.4E−10	2.45222	up	TG	C00422	
TG 14:0_15:0_16:1	778.625	1.6E−13	2.40301	up	TG	C00422	
TG 14:0_16:1_16:1	700.962	3.6E−05	2.42613	up	TG	C00422	
TG 14:1_16:1_16:1	606.393	1.4E−08	2.36593	up	TG	C00422	
TG 16:0_16:1_16:2	439.382	3.6E−06	2.32263	up	TG	C00422	
TG 10:0_16:0_16:1	364.977	3.3E−05	2.45687	up	TG	C00422	
TG 14:1_16:1_17:1	301.461	8.2E−11	2.25338	up	TG	C00422	
TG 15:0_16:0_16:1	260.848	0.00053	2.25966	up	TG	C00422	
TG 16:1_16:1_17:1	257.263	1.7E−11	2.20373	up	TG	C00422	
MGDG O-16:2_5:0	0.07153	1.3E−05	1.58418	down	EtherMGDG	–	
SMGDG O-11:0_28:6	0.06672	1.6E−08	1.6519	down	EtherSMGDG	–	
PE 38:4	0.06231	3.2E−05	1.65122	down	PE	C00350	
LDGCC 40:6	0.06065	1.2E−11	1.67241	down	LDGCC	–	
PC O-18:1_18:1	0.05768	3.1E−08	1.70446	down	EtherPC	C05212	
LPC 36:2-SN1	0.0561	4.8E−06	1.71966	down	LPC	C04230	
PE O-18:2_22:5	0.05108	5.2E−07	1.74204	down	EtherPE	C04475	
CAR 16:0	0.03533	1.8E−05	1.787	down	CAR	C02301; C02990	
CAR 18:1	0.02986	6.1E−08	1.8646	down	CAR	C02301	
LPC 30:0-SN1	0.02456	1.6E−09	1.90293	down	LPC	C04230	
PS 18:0_22:6	0.01993	1.9E−05	1.82679	down	PS	C02737	

The differentially expressed lipids were analyzed for biochemical pathway enrichment using the KEGG database (Fig. 4). We found that differentially regulated lipid molecules in the mature adipocytes were broadly related to metabolism (e.g., ether lipid metabolism and SP metabolism) and organismal systems (e.g., adipocytokine signaling pathway and regulation of lipolysis in adipocytes).

Figure 4 Lipid metabolic pathway analysis of the identified differentially expressed lipid species.

Differential gene expression between undifferentiated and differentiated 3T3-L1 cells

Cell samples at D0 and D10 were analyzed via RNA-sequencing (RNA-seq). The main characteristics of the libraries are shown in Table S4. The libraries contained 49,519,629 raw reads on average. After removing adaptors and low-quality/ambiguous sequences, an average of 47,209,124 valid clean reads remained. Among all samples, 96.45% of the valid reads mapped to the mouse genome database, including 72.76% unique mapped reads and 23.69% multi-mapped reads (Table S5). The gene expression profiles of D0 and D10 samples were analyzed with PCA, which revealed significant differences in gene expression patterns between D10 and D0 (Fig. 5A). Volcano plots were used to visualize the distribution of differentially expressed genes (DEGs) between D10 and D0 cells. There were 6,193 DEGs between D10 and D0, including 1,878 up-regulated and 4,315 down-regulated genes (Fig. 5B). The heatmap comparison represents some gene expression changes (Fig. 5C).

Figure 5 RNA-seq analysis of undifferentiated and differentiated 3T3-L1 cells.

(A) PCA results of the two groups. (B) Volcano plot of DEGs in undifferentiated and differentiated 3T3-L1 cells. Down-regulated genes are represented by blue dots and up-regulated genes are represented by red dots. (C) Heatmap comparison represents gene expression changes.

We next validated the RNA-seq data with qRT-PCR for randomly selected genes. The relative expression levels of acyl-coenzyme A dehydrogenase medium chain (Acadm), acyl-CoA synthetase long-chain family member 1 (Acsl1), angiopoietin-like 4 (Angpt14), fatty acid binding protein 5 (Fabp5), lipase hormone sensitive (Lipe), patatin-like phospholipase domain containing 2 (Pnpla2), stearoyl-Coenzyme A desaturase (Scd1), and lipoprotein lipase (Lpl) were significantly increased in differentiated 3T3-L1 cells (D10, p < 0.05, Fig. 6). The expression trends for all genes validated by qRT-PCR were consistent with the results from RNA-seq analysis, demonstrating the high quality of the sequencing data.

Figure 6 Gene expression levels from RNA-seq analysis (FPKM) and qRT-PCR (relative expression).

qRT-PCR was used to analyze expression levels of Acadm, Acsl1, Angptl4, Fabp5, Lipe, Pnpla2, Scd1, Npy1r, Pik3cd, and Lpl. The 18s, B2m, and β-actin genes were used as internal references for standardization. Bar graph (Blue) showing results from qRT-PCR (left ordinate). The line chart represents the results from RNA-seq analysis (right ordinate in red).

The 6193 DEGs identified with RNA-seq were analyzed for functional enrichment using the Gene Ontology (GO) database. There were significantly enriched biological processes (e.g., lipid metabolic process, fatty acid biosynthetic and lipid transport), cellular components, and molecular functions (Fig. S1). Enrichment analysis was also conducted using the KEGG database (Fig. 7). The integrated DEGs of undifferentiated and differentiated 3T3-L1 cells were mainly enriched in thermogenesis, the PPAR signaling pathway, and regulation of lipolysis in adipocytes.

Figure 7 Bubble diagram of KEGG enrichment result.

Bubble color corresponds to the p value for statistical significance of KEGG pathway enrichment. Bubble size is proportional to the number of genes annotated in a particular pathway.

Joint analysis of RNA-seq and lipid metabolome results

We compared the enriched KEGG pathways between DEGs and differentially expressed lipid metabolites. There were seven pathways enriched in both datasets (p < 0.05), namely adipocytokine signaling pathway, AGE-RAGE signaling pathway in diabetic complications, glycerolipid metabolism, insulin resistance, oxidative phosphorylation, regulation of lipolysis in adipocytes, and retrograde endocannabinoid signaling (Fig. 8 and Table S6). Genes with a fold change >5 and FPKM >1 (Table 2) were strongly correlated with metabolites in these pathways (Fig. S2 and Table S7). All of the DEGs were screened and a protein interaction network was constructed using String. The key hub genes were identified with Cytohubba in Cytoscape. The top ten hub genes were Pnpla2, diacylglycerol O-acyltransferase 1 (Dgat1), diacylglycerol O-acyltransferase 2 (Dgat2), fatty acid binding protein 4 (Fabp4), adiponectin C1Q and collagen domain containing (Adipoq), Lipe, Acsl1, solute carrier family 2, member 4 (Slc2a4), Lpin1, and Lpl (Fig. 9).

Figure 8 Histogram showing p values for statistically significant KEGG pathway enrichment overlapping between the differentially expressed gene and differentially expressed metabolite datasets.

Table 2 Differentially expressed genes (fold change > 5 and FPKM > 1) in the seven pathways enriched in both datasets.

pathway_id	pvalue	Gene_name	fc	log2(fc)	
ko04920	0.03	Slc2a4	705739.12	19.43	
ko00190	0.00	Cox8b	155059.57	17.24	
ko04923	0.00	Aqp7	110123.95	16.75	
ko04920	0.03	Adipoq	9075.14	13.15	
ko04923	0.00	Plin1	1645.29	10.68	
ko04923	0.00	Fabp4	888.29	9.79	
ko04920	0.03	Cd36	788.90	9.62	
ko00190	0.00	Cox7a1	176.70	7.47	
ko04933	0.00	Agt	92.86	6.54	
ko00561	0.02	Dgat2	82.57	6.37	
ko04923	0.00	Lipe	75.99	6.25	
ko00561	0.02	Pnpla2	69.29	6.11	
ko04920	0.03	Acsl1	60.84	5.93	
ko00561	0.02	Agpat2	48.39	5.60	
ko00561	0.02	Gpat3	35.15	5.14	
ko04931	0.00	Nr1h3	29.12	4.86	
ko00561	0.02	Lpin1	27.66	4.79	
ko00190	0.00	Ppa1	25.97	4.70	
ko00561	0.02	Dgat1	20.42	4.35	
ko00561	0.02	Lpl	15.51	3.96	
ko00480	0.24	Mgst3	15.38	3.94	
ko04923	0.00	Plaat3	15.32	3.94	
ko04931	0.00	Ppargc1b	13.26	3.73	
ko04920	0.03	Adipor2	12.39	3.63	
ko00190	0.00	Cyc1	7.33	2.87	
ko00190	0.00	Uqcrfs1	6.62	2.73	
ko04923	0.00	Abhd5	5.86	2.55	
ko00190	0.00	Uqcr11	5.54	2.47	
ko00190	0.00	mt-Co1	5.18	2.37	
ko04931	0.00	Ppp1r3c	5.07	2.34	
ko04920	0.03	Cpt1a	0.20	−2.32	
ko04723	0.00	Gng11	0.20	−2.33	
ko04933	0.00	Smad3	0.17	−2.57	
ko04933	0.00	Mmp2	0.17	−2.60	
ko04933	0.00	Col3a1	0.16	−2.64	
ko04933	0.00	Jun	0.16	−2.65	
ko04933	0.00	Col4a5	0.16	−2.69	
ko04933	0.00	F3	0.11	−3.14	
ko04931	0.00	Creb3l1	0.11	−3.15	
ko04933	0.00	Vegfc	0.10	−3.27	
ko04933	0.00	Col1a2	0.08	−3.61	
ko04933	0.00	Egr1	0.08	−3.73	
ko04933	0.00	Col1a1	0.07	−3.94	
ko04920	0.03	Socs3	0.06	−4.05	
ko04933	0.00	Fn1	0.02	−5.38	
ko04933	0.00	Ccl2	0.02	−5.89	

Figure 9 Interaction network of the top 10 hub genes.

Expression patterns of hub genes and key genes that regulate lipid metabolism during adipogenesis

Expression patterns of the top ten hub genes were analyzed in cell samples collected at D0, D2, D4, D6, D8, and D10 during adipogenesis. In addition, expression of the key adipogenesis genes CCAAT enhancer binding protein alpha (Cebp/ α) and peroxisome proliferator activated receptor gamma (Ppar γ) were also quantified; both were significantly activated during differentiation (Fig. S3). Pnpla2 (Fig. 10A), Dgat1 (Fig. 10B), Fabp4 (Fig. 10D), and Lipe (Fig. 10F) were significantly upregulated during differentiation, with the highest expression at D4. Dgat2 was highly expressed at D2 and D4 during differentiation, but the expression level was lower than in undifferentiated cells at D10 (Fig. 10C). Adipoq (Fig. 10E), Acsl1 (Fig. 10G), Lpin1 (Fig. 10H), Lpl (Fig. 10I), and Plin1 (Fig. 10J) were significantly upregulated during differentiation, with the highest expression at D8.

Figure 10 Relative expression of top 10 hub genes.

qRT-PCR was used to analyze expression of the top 10 hub genes. 18s, B2m, and β-actin served as the internal reference genes.

Discussion

In this study, we integrated lipidomics and RNA sequencing to reveal significant changes in lipid content and gene expression profiles between undifferentiated and mature 3T3-L1 cells during adipogenesis. Previous research examined the lipid change between undifferentiated and differentiated 3T3-L1 cells. However, only the changes of SM, PC, TG, PI, PE, and FA were analyzed (Popkova et al., 2020). Here, we found that in addition to the changes above mentioned, many other lipids showed significant changes after differentiation, such as LPC, EtherPE, and DG. Furthermore, many studies have conducted transcriptome sequencing during adipogenesis, and numerous DEGs related to the initiation of adipogenesis have been identified (Mikkelsen et al., 2010; Duteil et al., 2014; Al Adhami et al., 2015; Siersbæk et al., 2017; Romero et al., 2018). But the results are quite variable, which could be due to different sample batches, testing platforms, and data processing methods. So, we combined lipidomics and transcriptomic studies, to provide a better understanding of the molecular mechanism of adipogenesis.

As expected, compared with undifferentiated 3T3-L1 cells, there were larger TG and DG fractions, especially TGs, in differentiated 3T3-L1 cells. This is consistent with a previous study that found that adipocytes have a higher level of TGs than cells in the preadipocyte state (Popkova et al., 2020). Upon insulin stimulation, Slc2a4 moves to the cell surface and transports glucose from the extracellular milieu into the cell (Watson, Kanzaki & Pessin, 2004). GPAT3 has catalytic activity for a variety of saturated and unsaturated long-chain fatty acyl-CoAs, such as oleoyl-CoA, linoleoyl-CoA, and palmitoyl-CoA (Cao et al., 2006). During adipogenesis GPAT activity is increased by 30- to 100-fold (Coleman et al., 1978). AGPAT2 catalyzes the second step of TG synthesis (the glycerol phosphate pathway, the main synthesis pathway of TG), which is highly expressed in adipose tissues (Gale et al., 2006). Lipins are PAP enzymes that can catalyze the dephosphorylation of phosphatidate to DG in TG biosynthesis (Péterfy et al., 2001; Csaki et al., 2013). Lipin-1 plays a key role in adipose tissue PAP activity (Kok et al., 2012). In differentiating preadipocytes, Lipin-1 is required for normal expression of key adipogenesis regulating genes, including PPAR γ and C/EBP α, and for the synthesis of TG (Zhang et al., 2008). DGAT catalyzes DG to form, it has DGAT1 and DGAT2 two isoforms (Shi & Cheng, 2009). Functional DGAT is required for LDs in adipocytes (Harris et al., 2011). ACSL1 functions directly in activating fatty acid synthesis of triglycerides (Li et al., 2009). A previous study showed that when Acsl1 was overexpressed in mouse hearts, triglyceride levels in cardiomyocytes increased by 12-fold (Chiu et al., 2001).The high expression of Slc2a4 indicated that cellular glucose transport capacity was enhanced during adipogenesis. We found upregulation of Gpat3, Agpat2, Lipin1, Dgat1, Dgat2, and Acsl1 during adipocyte differentiation, suggesting that these genes transcriptionally control TAG synthesis during the adipogenesis of 3T3-L1 cells (Fig. 11).

Figure 11 Regulation pathways of genes and lipid metabolites during 3T3-L1 adipogenesis.

The yellow circles represent the differentially expressed lipid metabolites, and red text represent differentially expressed genes.

We identified 73 SPs with significantly different levels between undifferentiated and differentiated adipose cells. Among these SPs, the most drastic changes were in levels of SM and Cer_NS. De novo ceramide synthesis begins with the condensation of palmitate and serine to form 3-keto-dihydrosphingosine. 3-keto-dihydrosphingol is then reduced to dihydrosphingol, which is subsequently acylated by the enzyme (dihydroceramide) synthetase to produce dihydroceramide. The final reaction of ceramide formation is catalyzed by dihydroceramide desaturase. Ceramide can be further metabolized to other SPs, such as sphingomyelin (SM) and Cer_NS (Ramstedt & Slotte, 2002). Pnpla2 encodes adipose triglyceride lipase (ATGL) and Lipe encodes HSL in adipose tissue. ATGL and HSL are major enzymes that promote the decomposition of triacylglycerols (TGs) in mouse white adipose tissue. ATGL performs the first step in TG catabolism generating DG and fatty acids. DG is subsequently degraded by HSL and monoglyceride lipase (MGL) into glycerol and fatty acids (Zimmermann et al., 2004; Gao & Simon, 2007). LPL encodes lipoprotein lipase which catalyzes the hydrolysis of triglycerides (Pingitore et al., 2016). Fabp4 encodes the fatty acid binding protein that binds long chain fatty acids and retinoic acid, and delivers them to their cognate receptors in the nucleus (Prentice, Saksi & Hotamisligil, 2019). The high expression of Pnpla2, Lipe, and Lpl suggests that the level of intracellular free fatty acids will increase and can be used for synthesis of other lipids. Fabp4 can then transport these metabolites to specific sites for further synthesis of other lipids, such as ceramide and SM (Fig. 11).

We also found 214 GPs with significantly different levels between undifferentiated and differentiated cells. The GPs with the most differences were PI, PC, PE, LPE, LPC, ether-PE and ether-PC. De novo formation of PE and PC in eukaryotes occurs through several pathways. PE can be synthesized through the cytidine diphosphate (CDP)-ethanolamine branch of the Kennedy pathway, whereas PC can be synthesized through the CDP-choline branch of the Kennedy pathway (Kennedy & Weiss, 1956) or through methylation of PE (Bremer, Figard & Greenberg, 1960). Biosynthesis of PI is catalyzed by phosphatidylinositol synthase (PIS), which produces phosphatidylinositol and cytidine-monophosphate from the substrate molecules inositol and CDP-diacylglycerol (Bankaitis & Grabon, 2011). Phospholipase A2 can convert PC and PE into LPC and LPE, respectively. (Makide et al., 2009; Liu et al., 2017). Plaat3 has phospholipase A1 and A2 activity (Mardian, Bradley & Duncan, 2015), and can therefore catalyze the release of fatty acids from glycerophospholipids (in the sn-1 or sn-2 position) to form lysophospholipid (Pang et al., 2012). Our lipidomics analysis showed that production of many DGs (e.g., DG 16:0_17:1 and DG 15:0_16:0) was increased during adipogenesis. The increased DG abundance could provide more sources for the synthesis of PI, PC, PE, ether-PE, and ether-PC. High expression of Plaat3 (Table 2) may be related to enrichment of LPC and LPE.

Conclusions

In the process of adipogenesis in 3T3-L1 cells, a series of genes, including Slc2a4, Gpat3, Agpat2, Lipin1, Dgat1, Dgat2, and Acsl1, were activated, resulting in large amounts of accumulated TG. In addition to TG synthesis regulated genes, other lipid-regulated genes, including Pnpla2, Lipe, and Lpl were also up-regulated, leading to changes in the content of other lipids such as SM, Cer, PI, PC, PE, etherPE, etherPC, PLC and PLE. This study provides a reference for understanding the mechanism of human obesity development, and a basis for finding effective ways to prevent obesity.

Supplemental Information

Supplemental Information 1 Supplementary Figures

Click here for additional data file.

Supplemental Information 2 Supplementary tables

Click here for additional data file.

Additional Information and Declarations

Competing Interests

Author Contributions

DNA Deposition

Data Availability

The authors declare there are no competing interests.

Yangli Pei conceived and designed the experiments, performed the experiments, analyzed the data, prepared figures and/or tables, and approved the final draft.

Yuxin Song performed the experiments, analyzed the data, prepared figures and/or tables, and approved the final draft.

Bingyuan Wang analyzed the data, authored or reviewed drafts of the paper, and approved the final draft.

Chenghong Lin performed the experiments, prepared figures and/or tables, and approved the final draft.

Ying Yang performed the experiments, authored or reviewed drafts of the paper, and approved the final draft.

Hua Li and Zheng Feng conceived and designed the experiments, authored or reviewed drafts of the paper, and approved the final draft.

The following information was supplied regarding the deposition of DNA sequences:

The RNA sequencing data is available at SRA: PRJNA795061.

The following information was supplied regarding data availability:

The raw measurements are provided in the Table S8.

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
