# Peer review of "Integrated lipidomics and RNA sequencing analysis reveal novel changes during 3T3-L1 cell adipogenesis"

_PeerJ, doi:10.7717/peerj.13417_

## Round 0.1 · original submission · Minor Revisions

All reviewers recognize the importance of your research, but also point out many points that need to be corrected. I think their comments are reasonable and will improve your manuscript. Please correct the manuscript according to their comments.

Reviewer 1 ·

Basic reporting

Authors carried out in-depth study focusing on the lipidomic and transcriptional changes during maturation of 3T3-L1 cells. The study is straightforward yet important, deepening our understanding of adipogenesis, and could thus stimulate future studies to unveil obesity related disease in human. Most of my concern is regarding the formatting of the manuscript. Authors should try to address the following comments and check thoroughly for any similar errors.

Experimental design

no comments

Validity of the findings

no comment

Additional comments

57: “The medium containing with DMEM” does not make sense. I suppose authors used DMEM supplemented with FBS et ac. Please change. Also please specify the manufacturer of materials used in the study.
68: microscope manufacturer.
70: could authors specify how many cells were used for each sample?
112: “by using” is a redundant expression. Remove by or using
116: what is MS2 metabolites?
127: MS2 or M2S?
128: “301 metabolites were up”: readers will understand what the authors mean, but this is not a proper expression. Same as the rest of this paragraph. Authors could use increased and decreased instead of up and down.
201: please define slc2a4 where it appears the first time in the experiment. Please check the whole ms for errors like this.
257: please state the full name of the cell line.
261: Authors should be more confident about their study. It will provide basic knowledge for interested readers in related subjects for sure. Lease remove “may”.
Discussion: authors should discuss briefly how the present study might benefit future studies in relation to obesity development, prevention, and treatment of obesity-induced diseases.
Figure1: scale bar should be added to each image.

·

Basic reporting

The manuscript 'Integrated lipidomics and RNA sequencing analysis reveal novel changes during 3T3-L1 cell adipogenesis' from Yangli Pei et al systematically analyzed the molecular changes from preadipocyets to mature adipocytes. The finings of this manuscript are foundamentally important to reveal the the underlying mechanisms for adipogenesis. I have several comments mainly for the figures presented.

Experimental design

Good experiment design to address a basic but essential question regarding adipogenesis.

Validity of the findings

1. Figure 1: Scale bars are missing from cell culture pictures. And did the D8 and D10 pictures had the same magnitude?
2. According to Figure 4, the biochemical pathway enrichment for differently expressed lipids indicates minor effects on regulation of lipolysis in adipocytes. But in contrast, authors wrote from line 145-148 that differently regulated lipids molecules in adipocytes related to" regulation of lipolysis in adipocytes". If lipolysis process indeed changes significantly in mature adipocytes, did the authors quantify the lipolysis between preadipocytes and mature adipocytes? or the genes expression for lipolysis?
3. Figure 6 was confusing to distinguish qPCR data from RNAseq data. In addition, statistical analysis is missing from this Figure.
4. Figure 7: KEGG enrichment results shows the pathway enrichment showed the thermogenesis pathway. This is very interesting, did the author checked the KEGG, what caused the significant increases in thermogenesis?

Reviewer 3 ·

Basic reporting

Actually,there are studies on the lipidome and transcriptome profile alterations during the adipocyte adipogenesis Zhang et al.(2018, Int J Mol Sci), Sun et al. (2020, Front Mol Biosci) and Popkova et al. (2020, Analytical and Bioanalytical Chemistry). However, it is meaningful to analyze the molecular mechanism of adipocyte differentiation by integrating these two omics.

Experimental design

1.There are some obvious mistakes that could have been eliminated by a simple revision of the manuscript. For example, spacing after the number (page9, 3.3, line155), number before the title (Materials and Methods 2.4-2.6), etc.
2.The author needs to provide the reference related to the the criteria for screening differential lipid molecules.
3. Because the number of samples in transcriptome and lipidomes analysis is different, it is important for the author to explain clearly how to carry out correlation analysis in section 2.6

Validity of the findings

1.line 116-line127, MS2 and M2S mean? Please confirm.
2.It is suggested to provide the magnification or scale of Oil red O staining pictures in Figure 1.

Additional comments

1.Put table1 in supplemental material.
2.It is suggested that the authors compare their results with other publised lipidome and transcriptome data of 3T3-L1 cells, and put them into the discussion.

---

## Round 0.2 · Minor Revisions

I checked your manuscript and found the problems to be revised before acceptance.
* * *
Comment from Reviewer 1
57: “The medium containing with DMEM” does not make sense. I suppose authors used DMEM supplemented with FBS et ac. Please change. Also please specify the manufacturer of materials used in the study.
Your Reply: We have revised this sentence, added the information of materials manufacturer (L64-69).
* * *
My suggestion: This revision is not good enough. DMEM is a medium itself, but not supplemented material. Therefore, the expression “The maintaining medium supplemented with DMEM (Gibico), ---” is incorrect, and should be revised correctly.
"DMEM containing 10%FBS (Gibco) and xxx was used to xxx".

The line numbers you entered in the response letter are different from the line numbers in any of the files you submitted. Please check the line number of the submitted file. Please find attached file for comparison.

I request to send the response letter again with the (correct) line number and the corrected statement.

---

## Round 0.3 · Minor Revisions

I read your manuscript carefully. And I found some problems to be revised before acceptance. I suggest minor revisions to this manuscript. Please see the attached file. The red font part is my suggestion.
I hope that you will make the appropriate revisions and resubmit your manuscript soon.

---

## Round 0.4 · accepted · Accept

I checked your revised manuscript and found that problems are solved. Congratulations.